# Unsupervised Anomaly Detection for IoT-Based Multivariate Time Series: Existing Solutions, Performance Analysis and Future Directions

**DOI:** 10.3390/s23052844

**Published:** 2023-03-06

**Authors:** Mohammed Ayalew Belay, Sindre Stenen Blakseth, Adil Rasheed, Pierluigi Salvo Rossi

**Affiliations:** 1Department of Electronic Systems, Norwegian University of Science and Technology, 7034 Trondheim, Norway; 2Department of Gas Technology, SINTEF Energy Research, 7034 Trondheim, Norway; 3Department of Mathematical Sciences, Norwegian University of Science and Technology, 7034 Trondheim, Norway; 4Department of Engineering Cybernetics, Norwegian University of Science and Technology, 7034 Trondheim, Norway

**Keywords:** anomaly detection, IoT, multivariate time series, sensor networks

## Abstract

The recent wave of digitalization is characterized by the widespread deployment of sensors in many different environments, e.g., multi-sensor systems represent a critical enabling technology towards full autonomy in industrial scenarios. Sensors usually produce vast amounts of unlabeled data in the form of multivariate time series that may capture normal conditions or anomalies. Multivariate Time Series Anomaly Detection (MTSAD), i.e., the ability to identify normal or irregular operative conditions of a system through the analysis of data from multiple sensors, is crucial in many fields. However, MTSAD is challenging due to the need for simultaneous analysis of temporal (intra-sensor) patterns and spatial (inter-sensor) dependencies. Unfortunately, labeling massive amounts of data is practically impossible in many real-world situations of interest (e.g., the reference ground truth may not be available or the amount of data may exceed labeling capabilities); therefore, robust unsupervised MTSAD is desirable. Recently, advanced techniques in machine learning and signal processing, including deep learning methods, have been developed for unsupervised MTSAD. In this article, we provide an extensive review of the current state of the art with a theoretical background about multivariate time-series anomaly detection. A detailed numerical evaluation of 13 promising algorithms on two publicly available multivariate time-series datasets is presented, with advantages and shortcomings highlighted.

## 1. Introduction

The paradigm Internet of Things (IoT) has enabled the recent widespread digitalization in a vast array of application domains. IoT is characterized by the pervasive deployment of smart and heterogeneous devices (e.g., sensors, actuators, RFIDs) interconnected through the Internet for direct communications without human intervention. Currently, there are more than 12 billion IoT devices and by 2030, the number of deployed IoT devices is expected to reach 125 billion [1]. Accordingly, a massive increase in the amount of data generated by IoT is realistic and expected to reach 79.4 zettabytes (ZB) by 2025 [2].

Industrial applications have heavily exploited the opportunities provided by the IoT, resulting in the Industry 4.0 revolution. Industry 4.0 is characterized by the integration of data, artificial intelligence (AI) algorithms, industrial machines and IoT devices to create an intelligent and efficient industrial ecosystem. Industrial IoT (IIoT) solutions have played a significant role in the digitalization of various industrial processes, including manufacturing and power generation. With IIoT and related digital technologies, advanced analytics and enhanced decision-support systems are in place leading to increased automation and efficiency [3].

IoT-gathered data are commonly used for monitoring and/or predicting the behavior of specific environments or systems of interest. They thereby enable, for example, device health assessments, anomaly detection for fault diagnosis and if–then analysis [4]. The data are typically structured as multivariate time series, which are ordered variable values sampled at regular or irregular time intervals. They describe the simultaneous temporal evolution of multiple physical or virtual quantities of interest. Thus, they represent the natural mathematical framework for describing and analyzing data collected from sensors in IoT systems. The time series may contain irregular patterns or anomalous features for various reasons, including system failures, sensor failures or malicious activities and detecting these patterns and features can be crucial for successful monitoring and prediction. Multivariate Time Series Anomaly Detection (MTSAD) is therefore an important research area and reliable detection methods are required for properly deploying digital technologies, especially for safety-critical applications.

The joint analysis of temporal patterns and measurement dependencies across different sensors (often exhibiting complex behavior) makes MTSAD a challenging problem. Additionally, labeling massive volumes of IoT-generated data is practically impossible in many real-world scenarios; thus, the need for effective unsupervised approaches. Unsupervised MTSAD has become extremely relevant in the era of Industry 4.0 for monitoring and predictive maintenance. To be labeled robust and effective, an unsupervised MTSAD algorithm must yield high recall for tolerable false alarm rates; detect (possibly complex) anomalies in high-dimensional, co-dependent data; and be noise-resilient [5].

### 1.1. Related Work

Over the past few years, numerous anomaly detection techniques have been developed in a wide range of application domains. The approaches can be model-based, purely data-driven or hybrid analytics, each with unique advantages and disadvantages and applied to different sensor measurements, such as images, videos and time-series datasets. With the continuous generation of massive IoT data, data-driven machine learning techniques are employed to detect anomalies in multi-sensor IoT systems. Different learning tasks are implemented in a supervised, semi-supervised or unsupervised mode. Unsupervised MTSAD encompasses a wide range of techniques, from conventional statistical and machine learning approaches such as time series autoregression, distance-based methods, clustering-based techniques, density-based methods and one-class classification models to advanced learning methods including Autoencoders, convolutional networks, recurrent networks, graph networks and transformers. For massive multivariate sequence datasets, the performance of conventional statistical and machine learning approaches in detecting anomalies is sub-optimal [6]. Recently, deep learning algorithms have been utilized for time series anomaly detection and have achieved remarkable success [5]. Deep learning approaches rely on deep neural networks (DNN) and are effective at learning representations of complex data such as high-dimensional time series and unstructured image, video and graph datasets. The automatic feature learning capability of DNNs eliminates the need for feature engineering by domain experts. It is thus effective in unsupervised large-scale anomaly detection.

The present work complements an array of earlier surveys and analyses of anomaly detection techniques. Chandola et al. [7] provide a comprehensive review of the conventional anomaly detection methods in a variety of domains such as intrusion detection, fraud detection and fault detection. Chalapathy et al. [6] present an overview of deep learning-based approaches, including Autoencoders, recurrent neural networks and convolutional neural networks. Cook et al. [8] provide an overview of IoT time series anomaly detection with a discussion on major challenges faced while developing an anomaly detection solution for the dynamic systems monitored by IoT systems. Pang et al. [5] also provide a recent survey on deep anomaly detection with a comprehensive taxonomy with the underlying assumptions, advantages and disadvantages of different DNN architectures. Erhan et al. [9] review state-of-the-art anomaly detection methods in the specific area of sensor systems including cybersecurity, predictive maintenance, fault prevention and industrial automation. Choi et al. [10] provide a background on anomaly detection for multivariate time-series data and comparatively analyze deep learning-based anomaly detection models with several benchmark datasets. Sgueglia et al. [11] present a systematic literature review on the IoT time series anomaly detection including potential limitations and open issues. While most of the earlier related works on anomaly detection only offer a theoretical framework, a few incorporate performance analysis of different approaches. Garg et al. [12] present a systematic comparison of multivariate time series anomaly detection and diagnosis by proposing new composite metrics.

### 1.2. Motivation and Contribution

This paper has two main contributions:
A comprehensive, up-to-date review of unsupervised anomaly detection techniques for multivariate time series. This is motivated by the continuous, rapid progress in development of unsupervised MTSAD techniques. Moreover, in comparable existing reviews (cf. Section 1.1), we consider a broader range of techniques while focusing our attention on multivariate data from physical or soft sensors.A thorough performance analysis of 13 diverse, unsupervised MTSAD techniques, featuring two publicly available datasets. Both quantitative performance results and qualitative user-friendliness assessments are considered in the analysis.

The rest of the paper is organized as follows: Section 2 describes the mathematical framework to operate with multivariate time series; Section 3 provides a detailed overview of the various approaches to unsupervised MTSAD; a description of the datasets considered for the quantitative performance assessment and related performance metric is placed in Section 4; Section 5 discusses and compares the numerical performance of selected unsupervised MTSAD algorithms over the various scenarios; some concluding remarks are given in Section 6.

## 2. Multivariate Time Series

A multivariate time series is a multi-dimensional sequence of sampled numerical values, e.g., representing the collection of measurements from multiple sensors at discrete time instants. Such values are usually collected in matrix form. More specifically, for a system with *K* sensors, we define the system state at discrete time *n* as
(1)$$\begin{array}{c} x\left[n\right]={\left(x_{1}\left[n\right],x_{2}\left[n\right],\ldots ,x_{K}\left[n\right]\right)}^{t}\;, \end{array}$$
where ${\left(\cdot \right)}^{t}$ denotes the transpose operator. Thus, a multivariate time series related to *K* sensors and *N* discrete time instants can be represented as a matrix
(2)$$\begin{array}{c} X=\left(x\left[1\right],x\left[2\right],\ldots ,x\left[N\right]\right)=\left(\begin{array}{cccc} x_{1}\left[1\right] & x_{1}\left[2\right] & \cdots & x_{1}\left[N\right]\\ x_{2}\left[1\right] & x_{2}\left[2\right] & \cdots & x_{2}\left[N\right]\\ \vdots & \vdots & \ddots & \vdots \\ x_{K}\left[1\right] & x_{K}\left[2\right] & \cdots & x_{K}\left[N\right] \end{array}\right)\;, \end{array}$$
where the (n,k)th entry $x_{k}\left[n\right]$ represents the measurement sensed by the *k*th sensor at the *n*th time instant (in most cases, samples are taken at regular time intervals. If this is not the case, the sequence of values x is coupled with the sequence of sampling times $t={\left[t_{1},t_{2},\ldots ,t_{N}\right]}^{t}$). In the case of a single sensor (K=1), the time series is named univariate.

Anomalies in univariate time series are usually classified as point anomalies, contextual anomalies or collective anomalies. A point anomaly occurs when a single sensor measurement deviates significantly from the rest of the measurements (e.g., an out-of-range measurement); these anomalies are often caused by unreliable/faulty sensors, data logging errors or local operational issues. Contextual anomalies occur when sensor measurement is anomalous in a specific context only, i.e., the measured value could be considered normal on its own, but not in the context of the preceding/subsequent measurements. Finally, collective anomalies are characterized by a sub-sequence of sensor measurements behaving differently than other sub-sequences. Graphical representations of these anomaly categories are shown in Figure 1.

In the case of multivariate time series data, the relationship between the measurements across time domain and across different sensors is more complex and the classification of point, contextual and collective anomalies is less precise. Even if an observation is not exceptionally extreme on any one sensor measurement, it may still be considered an anomaly if it deviates significantly from the typical pattern of correlation present in the rest of the time series. Therefore, the combination of individual analysis for each constituent univariate time series does not provide a complete view on the possible anomalous patterns affecting a multivariate time series [10]. Anomaly detection methods for multivariate time series must consequently take into account all variables simultaneously.

When designing various unsupervised MTSAD methods, we assume that a multivariate time series $X_{\mathrm{train}}\in R^{K\times N}$ containing measurements under normal conditions is available for training, i.e., creating a representation of the system g{·} that is sufficiently accurate to capture how measurements are generated under regular operation. For performance evaluation, we assume that a multivariate time series $X_{\mathrm{test}}\in R^{K\times M}$, with M≪N, containing measurements both under normal and anomalous conditions is available for testing, i.e., the location of the anomalies is known. We denote $a={\left(a_{1},a_{2},\cdots ,a_{M}\right)}^{t}$ the vector identifying the locations of the anomalies, i.e., $a_{m}=1$ (resp. $a_{m}=0$) if an anomaly is present (resp. absent) at discrete time *m*. The goal of the training algorithm is to identify a representation such that $y=g\{X_{\mathrm{test}}\}$ with y being as close as possible to a according to some pre-defined metric.

## 3. Unsupervised MTSAD

The exponential growth of data generated by IoT devices makes manually labeling for anomaly detection infeasible in many practical applications. Thus, unsupervised anomaly detection solutions, which do not require any data labeling, are becoming increasingly relevant. In this section, we present and discuss a diverse selection of important unsupervised MTSAD strategies.

One possible way of categorizing the various unsupervised MTSAD methods is based on the underlying approach they use to detect an anomaly in the data. Most methods fall within one of the three approaches listed in Figure 2, i.e., reconstruction approach, prediction approach and compression approach.

In reconstruction-based approaches, the training multidimensional time series is compressed to a low-dimensional latent space and reconstructed back to its original dimension. These approaches are based on the assumption that anomalies are not well reconstructed, so we can use the reconstruction error or reconstruction probability as an anomaly score. The most common algorithms in this category are principal component analysis (PCA) and Autoencoders (AE).

In the prediction-based approach, we use current and past (usually from a finite-size sliding window) values to forecast single or multiple time steps ahead. The predicted points are compared to the actual measured values and an anomaly score is calculated based on the level of deviation: a significant difference in the observed and predicted data is considered anomalous. Vector autoregressive (VAR) models and Recurrent Neural Networks (RNN) are two common examples of such methods.

Compression methods, like reconstruction techniques, encode segments of a time series in a low dimensional latent space. However, instead of using the latent space to reconstruct the subsequences, the anomaly score is computed in the latent space directly. Dimensionality reduction methods reduce computation time and can also be used as a way to reduce model complexity and avoid overfitting. If the dimensions of the latent space are uncorrelated, compression methods also enable use of univariate analysis techniques without disregarding dependencies between variables in the original data.

Another common classification of unsupervised MTSAD methods utilizes the following three classes: (i) Conventional techniques, based on statistical approaches and non-neural network based machine learning; (ii) DNN-based methods, based on deep learning; and (iii) composite models, combining two or more methods from the previous categories into a unified model. Figure 3 shows the classification of relevant unsupervised MTSAD methods according to this classification, where the color of each method is selected according to the underlying approach previously discussed.

### 3.1. Conventional Techniques

Time series anomaly/outlier detection techniques have been used in statistics and machine learning for a long time. This section provides a brief summary of conventional time series anomaly detection methods that paved the way for more recent data-intensive methods.

#### 3.1.1. Autoregressive Models

A time series collected from a non-random process contains information about its behavior and suggests its potential future evolution. The autoregressive models are a type of regression model that uses past values to make predictions about the system’s future states. The autoregressive models can be used to detect anomalies by assigning an anomaly score based on the degree to which the actual value deviates from the predicted one. Autoregression (AR), moving average (MA), autoregressive moving average (ARMA) and autoregressive integrated moving average (ARIMA) models are the most common types of autoregressive models for univariate time series. More specifically, the AR(p) model is composed of a linearly weighted sum of *p* previous values of the series, whereas the MA(*q*) model is a function of a linearly weighted sum of *q* previous errors in the series. The ARMA(p,q) model incorporates both the AR and MA components and the ARIMA(p,d,q) model adds a time-difference preprocessing step to make the time series stationary. The difference order (*d*) is the number of times the data has to be differenced to make it stationary.

The vector autoregression (VAR) model is a common type of regressive model for multivariate time series. The VAR model is based on the idea that the current value of a variable can be represented as a linear combination of lagged values of itself and/or the lagged values of other variables, plus a random error term that accounts for all factors that the historical values cannot explain [13].

For a zero-mean multivariate time series {x[n]}, the VAR(p) model is: (3)$$\begin{array}{c} x\left[n\right]=A_{1}x\left[n-1\right]+A_{2}x\left[n-2\right]+\cdots +A_{p}x\left[n-p\right]+\varepsilon \left[n\right]\;,\;n=0,\pm \;1,\pm \;2,\ldots \end{array}$$
where each $A_{j}$ (j=1,2,…,p) is a matrix with constant coefficients and ε[n] is a multivariate zero-mean white noise. VAR models can predict multivariate time series by capturing the interrelationship between variables, making them applicable to multivariate time series anomaly detection tasks [14,15]. However, such regressive models are computationally expensive for huge datasets and thus less efficient for IoT anomaly detection.

#### 3.1.2. Control Chart Methods

Control charts are statistical process control (SPC) tools used to monitor the mean and variance shift of time series data. They are used to determine if the variation in the process is background noise (natural variations) or caused by some external factor. The presence of natural variations indicates that a process is under control, while special variations indicate that it is out of control. A control chart consists of a center line representing the average measurement value, the upper control limit (UCL) and the lower control limit (LCL) of the normal process behavior. The Shewhart *X*-chart, the cumulative sum (CUSUM) and the exponentially weighted moving average (EWMA) are the most frequently used control charts for determining if a univariate measurement process has gone out of statistical control [16]. For multivariate time series, the common multivariate control charts include Hotelling $T^{2}$ control chart, multivariate cumulative sum (MCUSUM) and multivariate EWMA (MEWMA) [17].

There are two stages in the control chart setting: Phase I focuses on the design and estimation of parameters (including control limits, in-control parameters and removal of outliers), while Phase II collects and analyzes new data to see if the process is still under control. Let X=(x[1],x[2],…,x[N]) represent a stream of *N* measurements from *K* sensors modeled as identical and independently random vectors according to a *K*-dimensional multivariate normal distribution with unknown mean vector μ and covariance matrix Σ, i.e., x[n]∼N(μ,Σ). The sample mean vector and the sample covariance matrix for *N* observations are computed as:(4)$$\bar{x}=\frac{1}{N}\sum_{n=1}^{N}x\left[n\right],\;S=\frac{1}{N-1}\sum_{n=1}^{N}\left(x\left[n\right]-\bar{x}\right){\left(x\left[n\right]-\bar{x}\right)}^{t},$$
respectively. Moreover, the $T^{2}$ statistic for the *n*th measurements is computed as follows [18]:(5)$$T_{n}^{2}=\left(x\left[n\right]-\bar{x}\right)S^{-1}{\left(x\left[n\right]-\bar{x}\right)}^{t},$$
and follows a Chi-squared distribution with *K* degrees of freedom. An anomaly is detected if $T_{n}^{2}$ is larger than a threshold related to the UCL [19]. MCUSUM uses a similar detection mechanism, but uses both past and current information to compute the test statistics [20]. Control charts usually requires an assumption of specific probability distributions of the data and may not be applicable to all datasets.

#### 3.1.3. Spectral Analysis

Spectral techniques attempt to approximate the data by combining attributes that capture most of the data variability. Such techniques assume that data can be embedded in a lower-dimensional subspace in which normal instances and anomalies appear significantly different [7]. Principal component analysis (PCA) and singular spectrum analysis (SSA) are two common methods in this category.

PCA is commonly employed to reduce data dimensions and improve storage space or computational efficiency by using a small number of new variables that are able to explain accurately the covariance structure of the original variables. Principal components are uncorrelated linear combinations of the original variables obtained via the singular value decomposition (SVD) of the covariance matrix of the original data and are usually ranked according to their variance [21]. Let C be a K×K sample covariance matrix calculated from *K* sensors and *N* measurements and denote $\left(e_{1},\;e_{2},\;\ldots ,\;e_{K}\right)$ and $\left(\lambda_{1},\;\lambda_{2},\;\ldots ,\;\lambda_{K}\right)$ the eigenvectors of C and the corresponding eigenvalues, respectively; then the *i*th principal component of the standardized measurements (Z) is computed as $y_{i}=e_{i}^{t}Z$. $Z={\left(z\left[1\right],\;z\left[2\right],\;\ldots ,\;z\left[p\right]\right)}^{t}$ is a vector of measurements defined as $z\left[i\right]=(x\left[i\right]-\bar{x})/\sigma$, where $\bar{x}$ and σ are sample mean and standard deviation, respectively.

Multivariate anomalies can be easily identified using PCA. One method involves evaluating how far each data point deviates from the principal components and then assigning an anomaly score [7]. The sum of the squares of the standardized principal components,
(6)$$\sum_{i=1}^{q}\frac{y_{i}^{2}}{\lambda_{i}}=\frac{y_{1}^{2}}{\lambda_{1}}+\frac{y_{2}^{2}}{\lambda_{2}}+\cdots +\frac{y_{q}^{2}}{\lambda_{q}},q\leq k$$
has a chi-square distribution with the degrees of freedom *q*. Given a significance level α, observation x is an anomaly if
(7)$$\sum_{i=1}^{q}\frac{y_{i}^{2}}{\lambda_{i}}>\chi_{q}^{2}\left(\alpha \right)$$
where $\chi_{q}^{2}\left(\alpha \right)$ is the upper α percentage in chi-square distribution and α indicates false alarm probability in classifying a normal observation as an anomaly.

SSA is a non-parametric spectral estimation method that allows us to identify time series trends, seasonality components and cycles of different period size from the original signal without knowing its underlying model. The basic SSA for 1D time series involves transforming the time series to the trajectory matrix, computing the singular value decomposition of the trajectory matrix and reconstructing the original time series using a set of chosen eigenvectors [22]. For multivariate time series, instead of applying SSA to each time series, multidimensional singular spectrum analysis (MSSA) is used [23]. When performing MSSA, we join the trajectory matrices of all time series, either horizontally (HMSSA) or vertically (VMSSA) and proceed to apply the same method [24].

SSA is a useful tool for detecting anomalies in time series due to its ability to separate principal components from noise in time series. SSA-based anomaly detection algorithms are based on the idea that the distance between the test matrix (obtained via target segmentation of time series) and the base matrix (reconstructed via *k*-dimensional subspace) can be computed using a series of moving windows [25]. A Euclidean distance (*D*) between the base matrix and the test matrix can be used as indicator for anomaly [26]. PCA anomaly detection requires assumption of specific distributions and SSA is computationally expensive for large IoT datasets.

#### 3.1.4. Clustering Algorithms

Clustering is a form of unsupervised machine learning in which data points are partitioned into a number of clusters with the goal of maximizing the similarity between points within the same cluster while minimizing the similarity between clusters. Maximum inter-cluster distance and minimum intra-cluster distance are two criteria for effective clustering [27]. A typical clustering anomaly detection algorithm consists of two phases: (i) time series data are clustered; (ii) dispersion of data points from their respective clusters is used to compute anomaly scores [28].

The *K*-means algorithm is among the most popular clustering approaches. The algorithm groups *N* data samples $\left({\left\{x\left[i\right]\right\}}_{i\;=\;1,\;\ldots ,\;N}\right)$ into *K* clusters, each of which contains roughly the same number of elements and is represented by a centroid $\left({\left\{\mu_{k}\right\}}_{k\;=\;1,\;\ldots ,\;K}\right)$. The optimal centroids can be determined by minimizing a cost function related to the intra-cluster variance [29]: (8)$$J=\sum_{k\;=\;1}^{K}\sum_{i\;\in \;C_{k}}||x\left[i\right]-\mu_{k}|{|}^{2}\;.$$

The algorithm is iterative and based on the following steps [30]: (i) cluster centroids are randomly initialized; (ii) data points are assigned to clusters based on their distance from the corresponding centroid (measured via Euclidean or other type of distance); (iii) centroids are updated by computing the barycenters of each cluster according to the assigned points; (iv) steps two and three are repeated until a stopping criterion is satisfied. Due to the unsupervised nature of this algorithm, the number of clusters must be specified in advance or determined using heuristic techniques. The elbow method is a popular choice for determining the number of clusters [31]: it involves plotting the cost function against different values of *K*. The point where the curve forms an elbow is selected as the optimum number of clusters (larger values provide negligible improvements). The *K*-means algorithm is prone to the cluster centroids initialization problem and does not perform well with multivariate datasets. PCA or other dimensionality reduction approaches are commonly employed as a form of pre-processing for reducing the impact of those issues [32]. Finally, anomalies are identified based on the distance from the centroids of the clusters [33,34].

The concept of medoid is used for the *K*-medoids clustering algorithm. Medoids are cluster representatives that have the smallest possible sum of distances to other cluster members. The *K*-medoids algorithm selects the center of each cluster among the sample points and is more robust to the presence of outliers, whereas the *K*-means algorithm does not (the barycenter of a set of sample points is not necessarily a sample point) and is very sensitive to outliers [35].

Density-Based Spatial Clustering of Applications with Noise (DBSCAN) [36] is a popular unsupervised clustering approach widely used for tasks requiring little domain expertise and relatively large samples. It operates on the principle that a point is part of a cluster if it is in close proximity to many other points from the same cluster and high-density regions are typically separated from one another by low-density areas. DBSCAN clusters multi-dimensional data using two model parameters: the distance that defines the neighborhoods (ϵ) and the minimum number of points in the neighborhoods (M). The ϵ-neighborhood of a point $x_{i}$ is defined as $N_{\epsilon }\left(x_{i}\right)=\left\{x\in D|dist\left(x_{i},x\right)\leq \epsilon \right\}$, where dist(·,·) is a distance function between two points and determines the shape of the neighborhood. Each sample point is categorized as a core point, boundary point or outlier based on these two parameters. More specifically, a point $x_{i}$ is a core point if its surrounding area with radius (ϵ) contains at least *M* points (including the point itself), i.e., when $|N_{\epsilon }\left(x_{i}\right)|\geq M$; a border point is in the ϵ-neighborhood of a core point but has fewer than *M* in its ϵ-neighborhood; a point is an outlier if it is neither a core point nor accessible from any core points. The DBSCAN algorithm can be implemented in the following four steps: (i) for each data point, find the points in the ϵ-neighborhood and identify the core points as those with at least *M* neighbors; (ii) create a new cluster for each core point, if that point is not already associated with one of the existing clusters; (iii) determine all density-connected points recursively and assign them to the same cluster as the core point; (iv) iterate through the remaining data points which have not been processed. The process is complete once all points have been visited and the points that do not belong to any cluster are considered noise. DBSCAN and its variants are commonly used in unsupervised anomaly detection [37,38] and rely on the assumption that points belonging to high-density (resp. low-density) regions are normal (resp. anomalous). When processing multivariate time series, DBSCAN treats each time window as a point, with the anomaly score being the distance between the point and the nearest cluster [39]. One-class support vector machine (OC-SVM) [40] is another data clustering algorithm that employ kernel ticks. OC-SVM separates normal training data from the origin by finding the smallest hyper-sphere containing the positive data. Clustering-based approaches for anomaly detection require high time and space complexity, but do not need any prior data labeling [28].

#### 3.1.5. Tree-Based Methods

Tree-based methods use tree structures to recursively split the data into non overlapping leaves and are particularly effective for high-dimensional and non Gaussian data distributions [41]. The Isolation Forest (IF) [42] and its variants are the most common methods in this category. IF is an ensemble unsupervised anomaly detection approach based on Decision Trees and Random Forests. In the IF algorithm, random sub-samples of data are processed in a binary isolation tree (also known as iTree) using randomly selected features. The samples that go further down on the iTree are less likely to be anomalies because they require more cuts to isolate them. Shorter branches indicate abnormalities because the tree can quickly isolate them from other data. The algorithm can be summarized in the following steps: (i) a random attribute $X_{A}$ and a threshold *T* between the minimum and the maximum value are selected; (ii) splitting the dataset into two subsets (if $X_{A}<T$), the point is assigned to the left branch; otherwise, it is sent to the right branch; (iii) recursively repeat the above steps over the dataset until a single point is isolated or a predetermined depth limit is reached. The process then recursively repeats steps (i) through (iii) to generate a number of Isolation Trees and, eventually yielding an Isolation Forest. The IF algorithm operates in a similar way to the Random Forest algorithm. However, the Random Forest algorithm uses criteria such as Information Gain to create the root node. The Isolation Forest algorithm is employed alone [43,44] or in combination with other techniques [45] to identify anomalies in sensor time series data. IF anomaly detection is based on the assumption that outliers will be closer to the root node (i.e., at a lower depth) on average than normal instances. The anomaly score is defined using the average depths of the branches and is given by the equation:(9)$$s\left(x,n\right)=2^{\frac{-E(h(x))}{c(n)}}$$
where c(n)=2H(n−1)−(2(n−1)/n). *n* is the sample size, h(x) represents the path length of a particular data point in a given iTree, E(h(x)) is the expected value of this path length across all the iTrees, H(i) is the harmonic number and c(n) is the normalization factor defined as the average depth in an unsuccessful search in a Binary Search Tree (BST). S(x,n) is a score between 0 and 1, with a larger value indicating a high likelihood of an anomaly [46]. IF is easy to implement and it is computationally efficient. However, it assumes individual sensor measurement are independent, which may not always be the case in real-world IoT data.

### 3.2. DNN-Based Methods

Recently, techniques based on deep learning have improved anomaly detection in high-dimensional IoT datasets. These approaches are capable of modeling complex, highly nonlinear inter-relationships between multiple sensors and are able to capture temporal correlation efficiently [5].

#### 3.2.1. Recurrent Networks

A common approach for DNN-based time series anomaly detection is the use of regression concepts to forecast one or more future values based on past values and then of the prediction error to determine if the predicted point is anomalous or not. Currently, several DNN-based series prediction models rely on Recurrent Neural Networks (RNN) [47], Long Short-Term Memory networks (LSTM) [48] and Gated Recurrent Units (GRU) [49]. RNNs are an extension of feed-forward neural networks with internal memory and are commonly used for modeling sequential data, such as time series, text and video. The RNN architecture consists of an input layer, a hidden layer and an output layer; however, unlike feed-forward networks, the state of the hidden layer changes over time. In the hidden layer, neurons are not only connected with the input layer and output layer but also with the neurons located in the same hidden layer. More specifically, the RNN input layer with *K* neurons receives a sequence of vectors (…,x[t−1],x[t],x[t+1],…) and the input units are connected to the hidden layer with *M* hidden units $h\left[t\right]={\left(h_{1},\;h_{2},\;\ldots ,\;h_{M}\right)}^{t}$ via a weight matrix $W_{ih}$. The recursive relation in the hidden layer (responsible for the memory effect) is expressed as:(10)$$h\left[t\right]=f_{h}\left(W_{ih}x\left[t\right]+W_{hh}h\left[t-1\right]+b_{h}\right)$$
where $f_{h}\left(\cdot \right)$ is the hidden layer activation function, $W_{hh}$ is a weight matrix defining the connection between the current and previous hidden state and $b_{h}$ is the bias vector in the hidden layer. The hidden state at time *t* is a function of the current input data and the hidden state at time t−1. The output layer with *L* units $y\left[t\right]={\left(y_{1},y_{2},\ldots ,y_{L}\right)}^{t}$ is determined by:(11)$$y\left[t\right]=f_{o}\left(W_{ho}h\left[t\right]+b_{o}\right)$$
where $f_{o}\left(\cdot \right)$ is the activation functions for output layer, $W_{ho}$ is weight matrix defining the connection between hidden units and output layer and $b_{o}$ is the bias vector in the output layer. RNNs are trained via a gradient descent approach referred to as backpropagation through time (BPTT) [50]; however, the exponential decay of the gradient makes RNN perform poorly when modeling long-range temporal dependencies (vanishing gradient problem) [51]. Recurrent networks such as LSTM and GRU have been introduced to avoid those problems affecting RNNs by utilizing different gate units to control new information to be stored and overwritten across each time step [52].

The LSTM architecture consists of a cell state and three control gates (input, forget and output gates). The cell state is the memory unit of the network and carries information that can be stored, updated or read from a previous cell state. The control gates regulate which information is allowed to enter the cell state. Input and forget gates regulate update/deletion of long-term memory retained in the cell state, while the output gate regulates the output from the current hidden state [48]. The internal operations of the LSTM cell are described by the following equations:(12)$$\begin{array}{cc} i[t] & =\sigma (W_{hi}h\left[t-1\right]+W_{xi}x\left[t\right]+b_{i}) \end{array}$$(13)$$\begin{array}{cc} f[t] & =\sigma (W_{hf}h\left[t-1\right]+W_{xf}x\left[t\right]+b_{f}) \end{array}$$(14)$$\begin{array}{cc} o[t] & =\sigma (W_{ho}h\left[t-1\right]+W_{xo}x\left[t\right]+b_{o}) \end{array}$$(15)$$\begin{array}{cc} \tilde{C}\left[t\right] & =\tanh(W_{hc}h\left[t-1\right]+W_{xc}x\left[t\right]+b_{c}) \end{array}$$(16)$$\begin{array}{cc} C[t] & =f\left[t\right]⊙C\left[t-1\right]+\left(1-f_{t}\right)⊙{\tilde{C}}_{t} \end{array}$$(17)$$\begin{array}{cc} h[t] & =o_{t}⊙\tanh\left(C_{t}\right) \end{array}$$
where i[t], f[t] and o[t] represent input, forget and output gates, respectively, $\tilde{C}\left[t\right]$ is the candidate cell state, C[t] is the cell state, h[t] is the hidden state and cell output, σ(·) is the sigmoid function, ⊙ is the Hadamard product, W is a weight matrix and b is the bias vector in each gate.

Recurrent networks are frequently utilized for time series anomaly detection tasks because of their prediction capabilities and temporal correlation modeling [53]. More specifically, the resulting prediction errors are assumed to follow a multivariate Gaussian distribution (with mean vector and covariance matrix usually computed via Maximum Likelihood Estimation), which is utilized to determine the probability of anomalous behavior. Telemanom is a framework based on standard LSTMs to detect anomalies in multivariate spacecraft sensor data [54].

#### 3.2.2. Convolutional Networks

Convolutional neural networks (CNNs) are feed-forward deep neural networks originally introduced for image analysis [55] and then used also for processing multidimensional time series and extracting correlations effectively. CNNs use convolution operations (in at least one layer) to extract patterns from the underlying (spatio)temporal structure of the time series. In comparison to fully connected networks, this often yields more efficient training and increased performance for similar model complexity.

A CNN typically consists of convolution, activation function, pooling and fully connected layers; each convolution layer consists of several filters whose values are learned via training procedures [56]. A window of multivariate time series is taken to create a matrix X and multiple filters of width *w* and height *h* (equal to the number of channels) are applied to generate multiple feature maps. In 1D convolution, the *k*th filter traverses in one direction on the input matrix X and outputs [57]:(18)$$h_{k}=f_{c}\left(W_{k}*X+b_{k}\right)$$
where $h_{k}$ is the *k*th output vector, ∗ represents the convolution operation, $f_{c}$ is the activation function and W and b are weight and bias, respectively.

Temporal Convolutional Networks (TCNs) are a variant of CNNs developed for sequential data analysis [58]. TCNs produce sequences by causal convolution, i.e., no information leakage from the future into the past. Modeling longer sequences with large receptive fields requires a deep network or a wide kernel, significantly increasing the computational cost. As a result, an effective TCN architecture employs dilated causal convolutions rather than causal convolutions, resulting in an exponentially increasing receptive field. DeepAnT [59] is a CNN-based approach developed to identify point and contextual anomalies in time series. The algorithm is a two-stage process, with the first step consisting of a CNN-based time series predictor that trains on the time series windows to make future predictions. The second phase involves an anomaly detector, which calculates the anomaly score from 0 to 1 based on the Euclidean distance between the predicted and actual values. Afterward, a threshold is set to identify normal and anomaly data. A TCN-based time series anomaly detection is employed in [60] with prediction errors fitted by a multivariate Gaussian distribution and used to calculate the anomaly scores. In addition, a residual connection is implemented to enhance prediction accuracy.

#### 3.2.3. Autoencoders

Multi-layer perceptron Autoencoders (MLP-AE) [47] is a type of unsupervised artificial neural network composed of sequentially linked encoder (*E*) and decoder (*D*) networks. The encoder maps the input vector $x\left[n\right]={\left(x_{1}\left[n\right],\;x_{2}\left[n\right],\;\ldots ,\;x_{K}\left[n\right]\right)}^{t}$ to a lower-dimensional latent code, $z\left[n\right]={\left(z_{1}\left[n\right],\;z_{2}\left[n\right],\;\ldots ,\;z_{L}\left[n\right]\right)}^{t}$ where L≪K and a decoder transforms the encoded representation back from the latent space to output vector $\hat{x}\left[n\right]={\left({\hat{x}}_{1}\left[n\right],\;{\hat{x}}_{2}\left[n\right],\;\ldots ,\;{\hat{x}}_{K}\left[n\right]\right)}^{t}$ that is expected to approximate the input vector x[n]. The input and reconstructed vectors are related via $\hat{x}\left[n\right]=D\left(E\left(x\left[n\right]\right)\right)$, where E(·) and D(·) denote the encoder and decoder operators, respectively, and the difference between the two vectors is called reconstruction error. AEs are trained to reduce the reconstruction error and usually the mean square error is the metric considered for the minimization procedure employed during the training process.

The conventional Autoencoders (AE)-based anomaly detection method is based on semi-supervised learning. The reconstruction error determines the anomaly score and samples with high reconstruction errors are considered anomalies. In the training phase, only normal data will be used to train the AE to identify normal data characteristics. During the testing phase, the AE will be capable of reconstructing normal data with minimal reconstruction errors. However, the reconstruction errors will be much higher than usual if the AE is presented with anomalous data (unseen before). An AE can determine whether the tested data are anomalous by comparing the anomaly score to a predefined threshold [61,62].

In conventional feedforward AEs, the two-dimensional spatial and temporal correlation are disregarded. To account for spatial correlation, more advanced reconstruction networks such as Convolutional AEs (CAEs) have been introduced [63]. CAEs exploit convolutions and pooling in the encoding stage, followed by deconvolution in the decoding stage. CAEs are frequently employed for image and video anomaly detection. In multivariate time series anomaly detection, the encoding process involves performing convolution operations on the input window of the time series, thus obtaining a low dimensional representation. The decoder process performs a deconvolution operation on latent representation to reconstruct the selected window: the convolution operation in CAEs generates spatial-feature representations among different sensors.

Variational AEs (VAEs) are another class of AEs which replace the reconstruction error with the reconstruction probability [64]. VAEs are unsupervised deep-learning generative methods that use Bayesian inference to model training data distribution and are based on three components: encoder, latent distribution and decoder. VAEs differ from conventional AEs due to a probabilistic interpretation for the anomaly score [65]. In VAEs, two conditional distributions are learned from the data to represent the latent variable given the observation in the original space, namely $q_{\phi }\left(z\left[n\right]|x\left[n\right]\right)$ and the observation in the original space given the latent variable, namely $p_{\theta }\left(x\left[n\right]|z\left[n\right]\right)$, where ϕ and θ are the set of parameters to be learned during training (e.g., weighting coefficients in case of neural networks are used for learning those distributions). The Kullback–Leibler divergence (or some approximated versions) is commonly used as the cost function to be minimized during the training procedure. In contrast to deterministic Autoencoders, VAE reconstructs the distribution parameters rather than the input variable itself. Consequently, anomaly scores can be derived from probability measures.

#### 3.2.4. Generative Models

Generative Adversarial Networks (GANs) are unsupervised artificial neural networks built upon two networks, respectively denoted generator (G) and discriminator (D), that are simultaneously trained in a two-player min-max adversarial game and are also commonly used in multivariate time series anomaly detection [66]. More specifically, the generator aims at generating realistic synthetic data, while the discriminator attempts to distinguish real data from synthetic data. The generator training goal is to maximize the likelihood of the discriminator making a mistake, whereas the discriminator training goal is to minimize its classification error, thus GANs are trained with the following objective function:(19)$$\min_{G}\max_{D}V\left(G,D\right)=E_{x\left[n\right]∼p_{data}}\left\{\log(D(x[n])\right\}+E_{z\left[n\right]∼p_{z}}\left\{\log(1-D(G(z[n]))\right\}$$
where V(·,·) is the value function of the two players.

GAN-based anomaly detection uses normal data for training and, after training, the discriminator is used to detect the anomalies based on their distance from the learned data distribution. Time Series Anomaly Detection with Generative Adversarial Networks (TAnoGAN) is a GAN-based unsupervised method that uses LSTMs as generator and discriminator models [67] while Multivariate Anomaly Detection with Generative Adversarial Networks (MAD-GAN) applies a similar framework to multivariate time series [68].

#### 3.2.5. Graph Networks

Graphs are a powerful tool for representing the spatial interactions among sensors; thus, they have been recently used for MTSAD by exploiting both inter-sensor correlations and temporal dependencies simultaneously. In a graph G=(V,E), each sensor is represented by a node/vertex v∈V and an edge e∈E models the correlations between two nodes [69,70].

Among graph-based techniques, graph neural networks (GNNs) generalize CNNs (defined over regular grids) by means of graphs capable of encoding irregular structures [71]. The use of GNNs for anomaly detection involves the identification of expressive representations on a graph so that normal data and anomalies can be easily distinguished when represented in the graph domains [72]. Graph Deviation Networks (GDNs) are a type of GNN that models the pairwise relationship via the cosine similarity of an adjacent matrix and then uses graph attention-based forecasting [73]. Multivariate Time-series Anomaly Detection via Graph Attention Network (MTAD-GAN) is a self-supervised graph framework that considers each univariate time series as an individual feature and includes two graph attention layers (feature-oriented and time-oriented) in parallel to learn the complex dependencies of multivariate time series in both temporal and feature dimensions [74]. MTAD-GAN jointly optimizes a forecasting-based model and a reconstruction-based model, thus improving time-series representations.

#### 3.2.6. Transformers

Modeling very long-term temporal dependencies and temporal context information is challenging for recurrent models such as RNN, GRU and LSTMs. Recently, transformers have achieved superior performances in many tasks of computer vision and speech processing [75]. Several time-series anomaly detection models have been proposed using transformer architectures: a reconstruction-based transformer architecture for early anomaly detection is proposed in [76]; a transformer-based anomaly detection model (TranAD) with an adversarial training procedure is proposed in [77]. Graph Learning with Transformer for Anomaly detection (GTA) has been studied to take advantage of both graph-based and transformer-based representations [78].

### 3.3. Composite Models

Hybrid algorithms combine deep-learning tools with classical approaches from statistics and signal processing. Some of the recent hybrid methods are briefly introduced below. RNNs have been combined with AEs, producing methods such as GRU-AEs and LSTM-AEs. EncDecAD is a method where both the encoder and decoder are based on LSTMs [79]. LSTM-VAEs combine LSTMs with VAEs [80]. OmniAnomaly proposes a stochastic RNN model for detecting anomalies in multivariate time series, arguing that random outliers can mislead deterministic approaches [81]. Multi-Stage Convolutional Recurrent and Evolving Neural Networks (MSCRED) combine convolution with an LSTM in encoder–decoder architecture in order to manage both spatial and temporal correlation [82]. Convolutional Long-Short Term Memory (ConvLSTM) networks [83] capture temporal patterns effectively in MSCRED by using the feature maps that encode inter-sensor correlations and temporal information. The Deep Autoencoding Gaussian Mixture Model (DAGMM) combines AE architecture with a GMM distribution for time series anomaly detection [84]. DAGMM optimizes the parameters of the deep AED and the mixture model simultaneously. UnSupervised Anomaly Detection (USAD) is a novel approach that employs AEs in a two-phase adversarial training framework [85] and overcomes the inherent limitations of AEs by training a model capable of identifying when the input data do not contain anomalies while the AE architecture ensures stability during adversarial training. Some composite models introduce a novel anomaly detection-based objective to build a model. Deep Support Vector Data Description (DeepSVDD) [86] introduces a one-class classification objective for unsupervised anomaly detection. It jointly trains a deep neural network while optimizing a data-enclosing hypersphere in output space.

## 4. Experimental Setup and Performance Metrics

In this section, we describe the datasets, data pre-processing, training hyperparameters, evaluation metrics, tools and selected methods.

### 4.1. Datasets

We have used two publicly available real-world multivariate time series datasets for our comprehensive performance analysis. Table 1 summarizes the features and statistics of each dataset.
(i)Secure Water Treatment (SWaT) dataset [87,88]: The SWaT system is an operational water treatment testbed that simulates the physical process and control system of a major modern water treatment plant in a large city. The testbed was created by the iTrust research unit at the Singapore University of Technology and Design. The SWaT multivariate time series dataset consists of 11 days of continuous operation, with 7 days collected under normal operations and 4 days collected with attack scenarios. Each time series consists of diverse network traffic, sensor and actuator measurements. For the last 4 days, a total of 36 attacks were launched. Actuators, flow-rate meters and water level sensors are all subjected to attacks, each with different intents and time frames. For our purpose, we re-sample the data by 1 min for computational efficiency.(ii)Server Machine Dataset (SMD) dataset [81]: SMD is a 5-week long (with 1-min sampling time) dataset acquired from a large Internet company. It comprises data from 28 servers, each monitored by 33 metrics. The testing set anomalies are labeled by domain experts. For our performance analysis, we used only one entity of the dataset.

### 4.2. Data Pre-Processsing and Tools

In our analysis, we performed downsampling, feature normalization and windowing of time series datasets. Downsampling speeds up neural network training and provides a denoising effect on the normal training data. A min-max scaling was considered for feature normalization and stable training of the models, i.e.,
(20)$$x^{\prime }=\frac{x-\min(X_{train})}{\max\left(X_{train}\right)-\min\left(X_{train}\right)}$$
where *x* is the actual measurement and $x^{\prime }$ is the value after scaling. For some of the algorithms, we utilized a sliding windowing of multi-sensor data to be used as input. Different window sizes and strides are selected for different datasets.

Both the Pytorch and Tensorflow deep learning frameworks are used to train and evaluate a selection of algorithms. In addition, the machine learning library Scikit-learn is used for performance analysis. Models are trained in the Google Colaboratory Pro environment using an NVIDIA T4 Tensor Core GPU processors.

### 4.3. Evaluation Metrics

For performance evaluations, we employed labeled test datasets and treated the anomaly detection problem as a binary classification task. Five different metrics were utilized to quantify the methods’ efficacy concerning this task: Precision, Recall, F1-score, Area under curve (AUC) and Area under Precision Recall (AUPR). These rely on the following parameters: the number of correctly detected anomalies, i.e., true positives (TPs), the number of erroneously detected anomalies, i.e., false positives (FPs) or false alarms, the number of correctly identified normal samples, i.e., true negatives (TNs) and the number of erroneously identified normal samples, i.e., false negative (FN). One of the most common performance metrics is the receiver operating characteristic (ROC), i.e., the relation between the true positive rate (TPR) and the false positive rate (FPR), defined, respectively, as
(21)$$TPR=\frac{TP}{TP+FN}=\frac{TP}{P}\;,\;FPR=\frac{FP}{FP+TN}=\frac{FP}{N}\;,$$
with P=TP+FN and N=FP+TN being the number of anomalous and normal samples, respectively. A related metric is the area under the curve (AUC) which provides the area bounded by the ROC on a (TPR,FPR) plane. Other common metrics, especially in the case of unbalanced scenarios (i.e., when the number of anomalous samples is much smaller than the number of normal samples), are precision, recall and F1-score ($F_{1}$), defined, respectively, as
(22)$$\mathrm{Precision}=\frac{TP}{TP+FP}=\frac{1}{1+\frac{N}{P}\frac{FPR}{TPR}}\;,\;\mathrm{Recall}=TPR\;,\;F_{1}=\frac{2\cdot \mathrm{Precision}\cdot \mathrm{Recall}}{\mathrm{Precision}+\mathrm{Recall}}\;.$$

A related metric is the area under the precision–recall (AUPR) curve which provides the area bounded by the precision–recall curve on a (P,R) plane.

## 5. Numerical Results and Discussion

In our numerical experiments, we conducted extensive tests on 13 different MTSAD algorithms covering all three of the main MTSAD categories discussed in Section 2. The algorithms are PCA, IF [42], OC-SVM [40] (conventional methods); VAE [64], MLP-AE [62], CNN-AE [89], GRU [49], LSTM [48], LSTM-AE [90], MAD-GAN [68] (DNN-based methods); ConvLSTM [83], USAD [85] and DAGMM [84] (composite models). For PCA, the anomaly score is the weighted Euclidean distance between each sample and the hyperplane formed by the specified eigenvectors. The IF algorithm uses a 100-tree ensemble as its estimator, splitting at a single node with the help of a single feature. The training data are sampled without replacement and used to fit individual trees. A polynomial kernel with a degree of 5 is utilized for the OC-SVM. Both MLP-AE and VAE are built from a three layer encoder and three layer decoder. The input channels are reduced to a 16 dimensional latent space vector and trained to minimize the mean square loss for 100 epochs. For CNN-AE, a one-dimensional convolution is applied with a kernel size of 5 and reduced from 128 filters to 32 filters in latent space. A multivariate window of time series is created with sequence length of 32 and 96.875% overlap. A look back of 120 observations is used to predict one step ahead for GRU, LSTM and ConvLSTM models. This means that the model takes the past 120 time steps as inputs and predicts the value for the next time step based on this historical information. Both GRU and LSTM have a similar architecture, consisting of three layers of stacked cells, where each cell has 64 neurons. The three layers of cells in a stacked RNN provide the network with the ability to learn complex temporal dependencies in the input data. For ConvLSTM, two sub-sequences with 60 time steps are created to apply convolution before using the result as input to LSTM cells. For backpropagation, a mini-batch gradient descent with different batch size is utilized and the adaptive moment estimation (ADAM) [91] optimizer is applied with a learning rate of $10^{-3}$. An early stopping criterion is set using a validation split of 5%. In the hidden layer, the activation function utilized is the Rectified Linear Unit (ReLU), whereas for the output layer, a Sigmoid function is employed. To avoid the overfitting problem, L2 regularization is used.

The performance of these algorithms is evaluated on both the SWaT and SMD datasets (and presented in Section 4.1). For each method, we report precision, recall, F1-score, AUC and AUPR, as defined in Section 4.3. The results are shown in Table 2 and Table 3 for the SWaT and the SMD datasets, respectively, where the highest scores for each metric are underlined. The ROC and precision–recall curves for the SWaT (resp. SMD) dataset are shown in Figure 4 (resp. Figure 5). We highlight that both the SWaT and SMD test datasets are significantly imbalanced. Therefore, F1-score and AUPR should be given more emphasis than AUC, since the latter may indicate artificially high performance for imbalanced datasets.

For the SWaT datasets, it is apparent that both conventional and DNN-based methods show comparable precision and recall. The prediction-based approaches such as LSTM, GRU and ConvLSTM show better performance than the Autoencoders (MLP-AE, CNN-AE and LSTM-AE). ConvLSTM, a method for simultaneously modeling spatio-temporal sequences outperforms other methods in terms of AUC (0.863) and F1-score (0.782), while also being among the best performing methods in terms of PRC (0.765). This is likely due to the high number of collective anomalies in the SWaT dataset, which ConvLSTM is especially suited for handling thanks to its combination of convolutions and temporal memory. The LSTM approach performs better in terms of AUPRC (0.777). This is because LSTM architecture is specifically designed to handle long-term dependencies and complex patterns in sequential data which allows it to be suited for collective anomalies present in the SWaT dataset.

The conventional approaches perform quite similarly, with IF showing high performance. IF, an ensemble classification method, also performs quite well compared to Autoencoders and has the higher AUC (0.825) and AUPRC (0.766) among the considered Autoencoder methods. Autoencoders, on the other hand, have a number of hyperparameters that can be tweaked to improve performance to the level of more traditional approaches. Among the DNN-based group, LSTM-AE has the lowest F1-score (0.712) and ConvLSTM the largest (0.782). This shows even complex neural reconstruction-based approaches are less efficient in identifying patterns in multivariate time series. Composite models such USAD and DAGMM do not perform well for SWaT, but better hyperparameter tuning could improve the results. We note that MAD-GAN is unstable during training and its performance depended sensitively on the hyperparameters used. The performance of USAD is also affected by the unstable adversarial learning of the Autoencoder architecture.

Although some methods (in particular ConvLSTM and LSTM) perform better than the others on the SWaT dataset, we highlight that all the methods perform “in the same ballpark”. In fact, the difference between the largest and smallest AUPR (resp. AUC) is less than 0.17 (resp. 0.06). Moreover, the ROCs and PRCs in Figure 4 all follow a similar pattern. It appears that almost all the methods are able to correctly classify around 60% of the anomalies in the test set without any false positives. However, as the anomaly threshold is reduced further to increase the recall/TPR, this results mostly in an increase in the amount of false negatives. That is, the precision decreases while the recall is more or less constant. The most likely explanation is that the SWaT dataset contains a significant amount of anomalies that are collective anomalies that are not outliers in the context of point anomalies. Our belief is that these are complex contextual anomalies and anomalous correlations between variables. In any case, it is clear that there is still significant room for improving upon the current state of the art performance. This motivates future development of novel MTSAD methods.

For the SMD dataset, low precision and high recall is observed due to the small amount of anomalies in the test dataset. DNN-based approaches perform reasonably well compared to the conventional methods. Again, ConvLSTM outperforms other techniques in terms of AUC (0.943) and AUPR (0.593) and also F1-score (0.606). This demonstrates that the approach is effective in capturing the temporal and spatial dynamics of multivariate time series. For the SMD dataset, the MAD-GAN algorithm exhibits the lowest performance with F1-score (0.173), AUC (0.852) and LSTM-AE with AUPR (0.318). We notice that the GAN-based methods generally perform poorly, but still show very high recall. This indicates that their discriminators have not been able to capture all the characteristics of normal data.

We conclude this section with some considerations related to the computational complexity and user-friendliness of the various algorithms previously considered. The tradeoff between computational cost and performance is a critical factor when selecting a MTSAD algorithm for IoT time series. From our analysis, we note that the GAN-based and composite architectures generally require large training/testing time compared to conventional methods. Kernel-based approaches, such as OC-SVM, also have heavy requirements in terms of training/testing time. ConvLSTM is slightly worse than average in terms of training time (157.8 s for SWaT and 1192 s for SMD), but better than average in terms of testing time (3.2 s for SWaT and 6.2 s for SMD). Among the considered methods in this paper, GAN-based methods were the least user-friendly: GANs are notoriously difficult to train due to the necessity of matching the discriminator and the generator to avoid saturation of the adversarial cost function. It is worth mentioning that conventional methods like PCA and IF are extremely user-friendly and performed reasonably well for both datasets. In contrast, training composite models takes longer time due to their complex architecture. Overall, as for the experiments reported in this work, ConvLSTM offered the best tradeoff when considering user-friendliness, computational complexity and performance. We experienced the method as stable during training and not critically sensitive to hyperparameters. Moreover, it achieved the highest AUPR score for both datasets.

## 6. Conclusions and Future Work

In this paper, we provided a comprehensive review of unsupervised Multivariate Time Series Anomaly Detection (MTSAD) methods. Massive volumes of unlabeled data, generated by IoT devices and sensors in the form of multivariate time series, capture either normal or anomalous behavior of monitored systems/environments. Several applications in various domains (industrial is one of the most relevant) require unsupervised MTSAD to be available and effective. We categorized unsupervised MTSAD techniques into three broad classes (according to the mechanism for outlier identification): reconstruction, prediction and compression. We further classified MTSAD approaches into three groups: (i) conventional approaches, which are based on statistical methods; (ii) deep learning-based methods; and (iii) composite models. Several methods in each group were described in detail; 13 specific techniques were selected for quantitative performance analysis and comparison using two public datasets; the most promising techniques were highlighted.

Despite the existence of several unsupervised MTSAD techniques in the current state-of-the-art, there are substantial challenges still open. Many methods are specific and tailored to particular use cases and no one-size-fits-all approach is available. Further research is needed to overcome the limitations of existing approaches and here we describe some of the promising directions.

Collective anomaly detection. Most of the previous unsupervised MTSAD algorithms primarily focus on point anomaly detection, while collective or sub-sequence time-series anomalies (more common in IoT-based systems) have been handled less frequently. Deep neural networks are expected to provide relevant improvements in this area.

Real-time anomaly detection. Although the capability to operate in (near) real-time is crucial for many use cases (e.g., IIoT, smart traffic and smart energy) involving short-term or automated decision-making, most existing methods lack the ability to detect anomalies efficiently in data streams. Addressing the challenge of evolving data streams in the context of IoT anomaly detection as well as related computational cost is crucial.

Irregular time series. Most unsupervised MTSAD techniques assume regular sampling of time-series data, which is not true in many real-world scenarios. Processing data to build a regular time series (e.g., using interpolation techniques) is not necessarily optimal; thus, detecting anomalies in irregular domains represents a relevant area for future research.

Explainable anomaly detection. Decision-support systems need to interact with human operators; thus, often the capability to provide a rationale for a decision is more relevant than the decision itself. Explainable Anomaly Detection (XAD), i.e., developing methods for anomaly detection coupled with related supporting motivation, is necessary for the methods themselves to be considered for final deployment in many relevant scenarios, e.g., safety-critical systems.

Hybrid models. Combining model-based and data-driven approaches into hybrid models is a relevant research direction aiming at maintaining the explainability/interpretability of the former and the accuracy of the latter. Complex non-stationary environments are scenarios in which such combination is expected to have a large impact.

Graph-based approaches. Graph neural networks (GNNs) and other graph-based methodologies are promising tools for dealing effectively with topology constraints in complex data. The investigation of how GNNs can be exploited for MTSAD anomaly detection is likely among the most promising directions for designing a new generation of IoT systems.

## Figures and Tables

**Figure 1 sensors-23-02844-f001:**
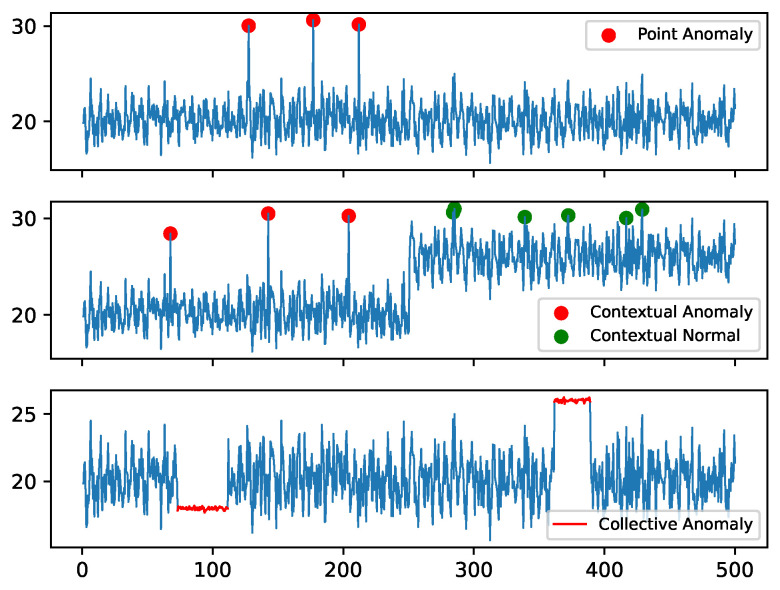
Types of anomalies.

**Figure 2 sensors-23-02844-f002:**
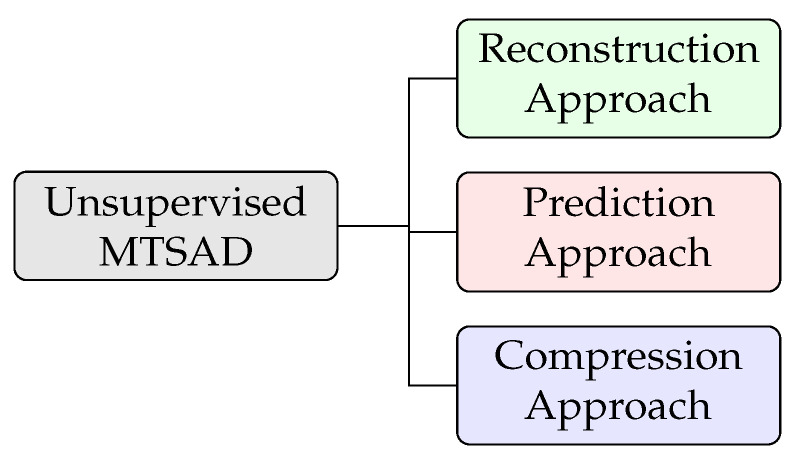
Approaches to unsupervised MTSAD.

**Figure 3 sensors-23-02844-f003:**
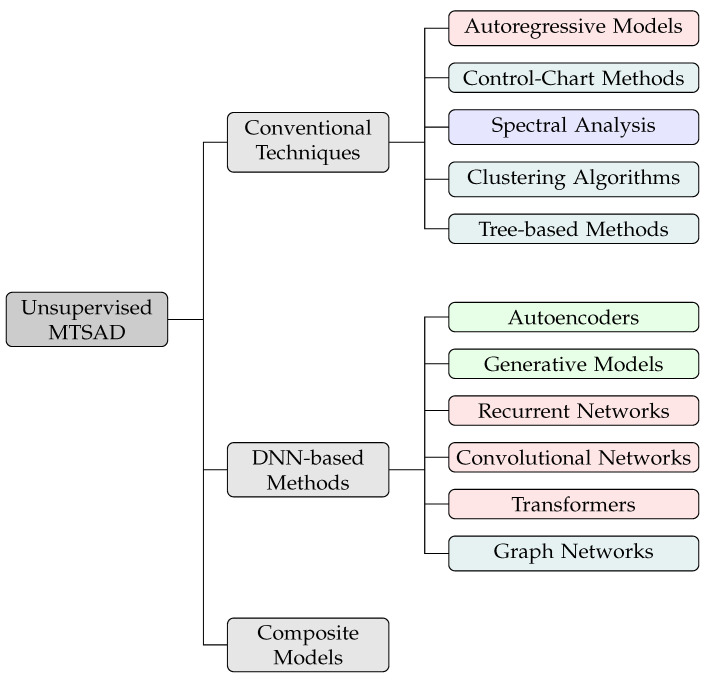
Unsupervised MTSAD methods. The colors red, green and blue indicate approach type as in Figure 2. Teal indicates that both compression and reconstruction approaches can be used.

**Figure 4 sensors-23-02844-f004:**
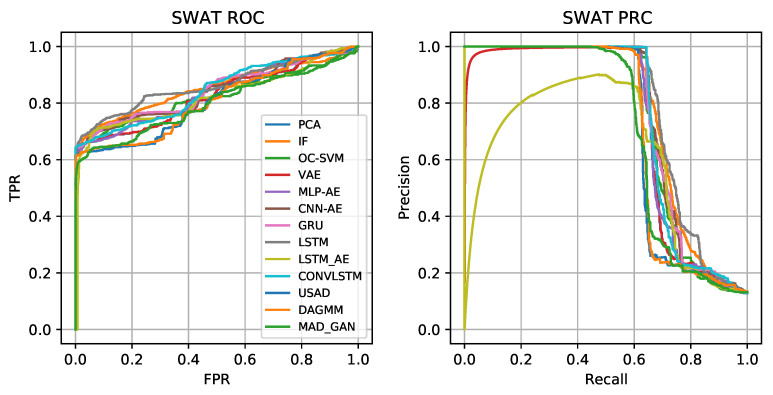
ROC and PRC for SWaT dataset.

**Figure 5 sensors-23-02844-f005:**
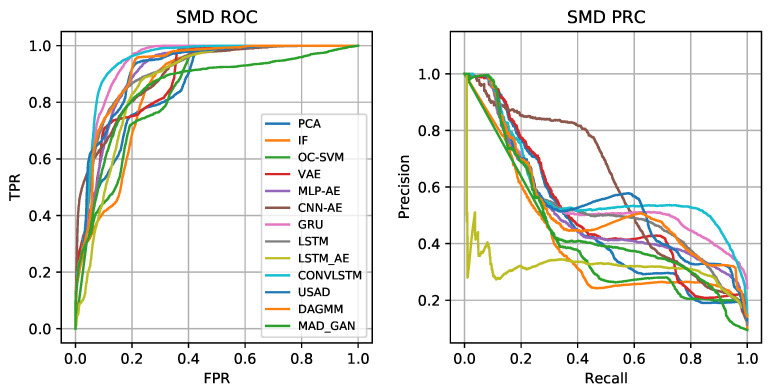
ROC and PRC for SMD dataset.

**Table 1 sensors-23-02844-t001:** Summary of the datasets.

Dataset	SWaT	SMD
No. of channels	51	38
Entities	1	28
Average Train size	495,000	25,300
Average Test size	449,919	25,301
Anomaly rate	12.140%	4.21%

**Table 2 sensors-23-02844-t002:** SWaT Results.

Method	Precision	Recall	F1-Score	AUC	AUPR	Train Time (s)	Test Time (s)
PCA	0.996	0.642	0.781	0.827	0.730	0.120	0.025
IF	0.998	0.617	0.762	0.854	0.766	0.752	0.217
OC-SVM	0.959	0.644	0.771	0.826	0.746	3.488	0.890
VAE	0.996	0.642	0.781	0.827	0.730	84.478	0.463
MLP-AE	0.996	0.620	0.764	0.836	0.738	29.342	0.489
CNN-AE	0.976	0.643	0.775	0.842	0.753	56.711	1.001
GRU	0.996	0.643	0.782	0.844	0.752	44.213	2.397
LSTM	0.998	0.643	0.782	0.862	0.777	35.060	2.812
LSTM-AE	0.856	0.610	0.712	0.822	0.604	33.244	2.067
ConvLSTM	0.998	0.643	0.782	0.863	0.765	157.828	1.824
USAD	0.989	0.614	0.758	0.808	0.706	269.967	3.274
DAGMM	0.971	0.614	0.752	0.807	0.707	226.149	2.896
MAD-GAN	0.912	0.589	0.716	0.801	0.700	682.528	1.947

**Table 3 sensors-23-02844-t003:** SMD Results.

Method	Precision	Recall	F1-Score	AUC	AUPR	Train Time (s)	Test Time (s)
PCA	0.399	0.489	0.439	0.861	0.477	0.107	0.045
IF	0.263	0.839	0.401	0.854	0.405	0.898	0.361
OC-SVM	0.281	0.714	0.403	0.844	0.408	71.983	27.043
VAE	0.424	0.699	0.528	0.883	0.510	261.104	1.742
MLP-AE	0.374	0.772	0.504	0.908	0.507	7.772	1.606
CNN-AE	0.475	0.605	0.532	0.900	0.607	152.477	3.185
GRU	0.454	0.785	0.576	0.937	0.568	178.168	7.267
LSTM	0.479	0.649	0.551	0.907	0.535	213.119	7.287
LSTM-AE	0.311	0.801	0.448	0.868	0.318	75.912	6.403
CONVLSTM	0.458	0.898	0.606	0.943	0.593	1192.153	6.245
USAD	0.095	1.000	0.173	0.915	0.549	969.609	15.762
DAGMM	0.095	1.000	0.173	0.918	0.533	826.505	13.248
MAD-GAN	0.095	1.000	0.173	0.852	0.465	2479.399	8.921

## Data Availability

The data presented in this study are available on the source mentioned in the text.

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
