# Peer review of "Unsupervised Anomaly Detection for IoT-Based Multivariate Time Series: Existing Solutions, Performance Analysis and Future Directions"

_sensors, 2023, doi:10.3390/s23052844_

Round 1

Reviewer 1 Report

The paper presented clearly  about multivariate time-series anomaly detection methods and should be added nomenclature table for readable easily.

Reviewer 2 Report

Unsupervised Anomaly Detection for IoT-based Multivariate Time Series: Existing Solutions and Performance Analysis

Review Report

Aim of the current work is to provide the extensive review of the current state of work in the direction of multivariate time series. Paper can be accepted after major revision. Some important issues are as follows:

1.      Statement in the abstract section “A detailed numerical evaluation of 13 promising algorithms on 2 publicly-available multivariate time-series datasets is presented, with advantages 15 and shortcomings highlighted.” not correlated with the fig.3. As in figure 3, 3 different bifurcations of Unsupervised MTSAD methods are provide. Clarify it.

2.      Further, the above statement is not correlated with the results provided in table2. In table 2, results are compared for 14 algorithms.  Clarify it.

3.      As this is the review paper, so while describing about Autoregressive Models, Control Chart Methods, Spectral Analysis, etc., authors must have to provide the chronological references from the beginning to the current year.

4.      Via fig.1, 3 different approaches to unsupervised MTSAD are presented graphically by the authors but they did not explain it theoretically.

5.      In conclusion section, provide some suggestions for future research work in this direction. 

Reviewer 3 Report

This paper analyses the performance of existing solutions regarding unsupervised Anomaly Detection for IoT-based Multivariate Time Series.

The paper is well-written and quite organized. However, I have the following points about it:

1. The authors must refer to Figure by Figure 1 instead of Fig.1, since the caption used Figure.

2.  The Contextual anomalies in Figure 1 are not clarified well, especially in the last sub-figure.

3. The paper is not cited well, several paragraphs in section 2 and section 3 are not cited.

4. In section 2, the transpose of (.), the period should be replaced by the vector.

5. Equation (20) should be referred to as normalization.

6.  The authors should refer to solutions (examined algorithms)((PCA, IF, OC-SVM), DNN-based methods (DeepSVDD, VAE, MLP- AE, CNN-AE, GRU, LSTM, LSTM-AE, MAD-GAN) and composite models (ConvLSTM, USAD, and DAGMM). ) by their references.

7. The evaluation metrics section should be reorganized.

8. the conclusion should clarify the achieved results and include some recommendations for the examined approaches.

Reviewer 4 Report

In this paper, the authors have performed an analysis of unsupervised multivariate time-series anomaly detection (MTSAD) methods. My comments are given as below.

* The contribution, motivation, and benefits of this research were missing. Please add in the introduction section which should be mentioned in bullets.

* The related work section was missing. Please add the most recent and state-of-the-art protocols in this section.

* Section 5 entitled numerical results and discussion is short. The details of Figures 4 and 5 and table 3 are required.

* The discussion and the implication of this research were missing. You should add the details in this section.

*The future work should be mentioned in the conclusion section.

Round 2

Reviewer 2 Report

NA

Author Response

Thanks for considering our work worth publication. 

Reviewer 4 Report

The authors modified the manuscript. However, it has not been fully addressed. Some minor comments are given below. Please address these comments in the final version.

 1. In the introduction section, the paper organization is missing. Please add; the paper organization is followed. Please have a look at the below references.

* Majeed, A. Mateen, R. Abbasi and S. O. Hwang, "A Systematic Survey on the Recent Advancements in the Social Internet of Things," in IEEE Access, vol. 10, pp. 63867-63884, 2022, doi: 10.1109/ACCESS.2022.3183261.

* W. -K. Lee, A. Mateen and S. O. Hwang, "Integration of Network science approaches and Data Science tools in the Internet of Things based Technologies," 2021 IEEE Region 10 Symposium (TENSYMP), Jeju, Korea, Republic of, 2021, pp. 1-6, doi: 10.1109/TENSYMP52854.2021.9550992.

2. Please do not rearrange or merge the introduction and related work section. There were two separate sections. The motivation, contribution, etc. should be the part of introduction section.

3. The related work should be a separate section for example; 2. Related work.

Author Response

Thanks for previous suggestions and for noticing the improvements of the paper with respect to the previous version. 

As for the new comments, we would like to mention that the paper organization was already present in the previous submitted version (it is found in Section 1.2, page 3, line 107 <<The rest of the paper is organized …>>); while we would prefer to keep the description of Related Work as a subsection of Section 1 (more precisely Section 1.1), since in our opinion it provides a better reading flow. We are aware that having a dedicated section for related work is a popular choice, but having it included as a subsection of the introduction is popular and accepted as well (e.g. D. Ciuonzo, P. Salvo Rossi "Distributed Detection of a Non-cooperative Target via Generalized Locally-Optimum Approaches," Information Fusion, Elsevier, Invited Paper at the Special Issue on Event-Based Distributed Information Fusion Over Sensor Networks, vol. 36, pp. 261-274, July 2017).